# Multi-objective AGV scheduling in an automatic sorting system of an unmanned (intelligent) warehouse by using two adaptive genetic algorithms and a multi-adaptive genetic algorithm

**Yubang Liu**[1◐]*, **Shouwen Ji**[1◐]*, **Zengrong Su**[2‡], **Dong Guo**[3‡]

**1** School of Traffic and Transportation, Beijing Jiaotong University, Beijing, China, **2** Aviation Business Department, Beijing capital international airport Company Limited, Beijing, China, **3** School of Mechanical-Electronic and Vehicle Engineering, Beijing University of Civil Engineering and Architecture, Beijing, China

◐ These authors contributed equally to this work.
‡ These authors also contributed equally to this work.
* swji@bjtu.edu.cn (SWJ); 17120842@bjtu.edu.cn (YBL)

**Data Availability Statement:** All relevant data are within the paper and its Supporting Information files.

## Abstract

Automated guided vehicle (AGV) is a logistics transport vehicle with high safety performance and excellent availability, which can genuinely achieve unmanned operation. The use of AGV in intelligent warehouses or unmanned warehouses for sorting can improve the efficiency of warehouses and enhance the competitiveness of enterprises. In this paper, a multi-objective mathematical model was developed and integrated with two adaptive genetic algorithms (AGA) and a multi-adaptive genetic algorithm (MAGA) to optimize the task scheduling of AGVs by taking the charging task and the changeable speed of the AGV into consideration to minimize makespan, the number of AGVs used, and the amount of electricity consumption. The numerical experiments showed that MAGA is the best of the three algorithms. The value of objectives before and after optimization changed by about 30%, which proved the rationality and validity of the model and MAGA.

## Introduction

The competition in the logistics industry has become increasingly fierce with the development of global e-commerce. As those who can provide logistics services more quickly will seize more market share, the emergence of intelligent warehouses and unmanned warehouses has been of great help. In these warehouses, some links, even the whole process, do not require manual participation. Therefore, logistics companies can achieve higher efficiency and solve the problem of recruitment to a certain extent [1]. As a result, the emergence of these warehouses is regarded as the gospel of logistics companies. Automated guided vehicle (AGV) is a logistics transport vehicle with high safety performance and powerful functions, which can genuinely achieve unmanned operation. The use of AGVs in the warehouse will make the

**Funding:** This research was supported by the National Key R&D Program of China (No. 2018YFB1601603). The financial support of this research has nothing to do with 'Beijing capital international airport Company Limited'.

**Competing interests:** Our paper has no commercial connection with Beijing capital international airport Company Limited, so this does not alter our adherence to PLOS ONE policies on sharing data and materials.

whole logistics process optimized, bringing a significant change to the logistics industry. Using AGVs that can work 24 hours a day in an automatic sorting system—the most time-consuming process in warehousing operations—will not only shortens the overall time used but also improves work efficiency. Therefore, a good AGV scheduling scheme will enable consumers' orders to be sorted and delivered to customers earlier, which will significantly enhance consumer experience and improve consumer satisfaction. Although there have been many studies on AGV scheduling, they are mostly concentrated in the manufacturing field, especially in the FMS [2–15]. In the logistics industry, the application of AGVs is still a new trend. Besides, the AGV scheduling remains an open research area due to the different facts to be considered in the scheduling process, for example, objectives, limitations, and considerations.

In the earlier works, the scheduling objective was mostly based on minimizing the makespan to ensure the operating efficiency of the multi-AGV system [2–5,10,11,15–17]. However, these studies ignored the impact of the number of AGVs used in the AGV system, for AGV is a relatively expensive device. If the number of AGVs used can be reduced, many costs will be saved [18–20]. Therefore, subsequent studies began to consider the number of AGVs to guide the actual scheduling while minimizing the makespan [21–23]. Moreover, another practical problem of AGV scheduling has been often neglected, namely the charging process of AGV. Most studies did not consider the battery charge of AGV, which was equivalent to assuming that AGV will not stop working even if its electric power is insufficient. Only in recent years, some studies have begun to add the quantity of electricity of AGV as a constraint to the scheduling considerations [14,24]. However, this is still not considered comprehensively, for the restriction of electricity is only to make AGVs in the system unable to complete tasks or sort orders when they are power off. In fact, they can be charged first and then go back to work. Consequently, a sound AGV scheduling system not only needs to have a job assignment but also should assign a charging task when the power of AGV is insufficient to meet the power demand of its subsequent work.

What is more important is that there is a common assumption in almost AGV scheduling studies that the AGV runs at a constant speed. However, the speed of the AGV is adjustable and controllable, so it is more practical to take the speed of the AGV into the consideration of the scheduling system. This consideration can give birth to a third objective—minimizing the amount of electricity consumed by all AGVs. The less power the system consumes, the fewer carbon emissions, which not only affects the cost of the operation but also takes responsibility for environmental protection.

The final decision is a trade-off among these three objectives (minimization of the makespan, the number of AGVs used, and the amount of electricity consumed). Minimizing the makespan means that more AGVs need to be arranged and those AGVs must be fast. The higher the AGVs' speeds are, the more electricity they consume per unit time. If the number of AGVs is minimized, the makespan will increase. The speed of the AGV will have to be as high as possible to complete all tasks, resulting in more electricity consumption. Minimizing the amount of electricity consumed by all AGVs requires that the AGV should not be too fast, which means that more AGVs will be needed, and the makespan will not be short. It is a practical and attractive study to find a balance between these three objectives.

To address these issues, considering the charge of the battery in AGV, the speed of the AGV, job assignment and charging task, a multi-objective mathematical model was developed to minimize the makespan, number of AGVs, and amount of electricity consumed by all AGVs to schedule AGVs in logistics sorting operations in this study.

Heuristic algorithm is the most commonly used method in AGV scheduling research, among which genetic algorithm (GA) is the most common [12, 25–33], and adaptive genetic algorithm (AGA) is an improvement of traditional genetic algorithm. The main idea of it is to

adjust adaptively the genetic parameters, which greatly improves the convergence accuracy of the genetic algorithm and increases the convergence speed. There are two kinds of adaptive genetic algorithms that are used frequently and perform quite well. The first is an adaptive genetic algorithm that generates adaptive crossover and mutation rate based on fitness values [34]; the second is based on the information entropy, especially the population entropy which is used to reflect population diversity [35]. We have noticed that some scholars have used the multi-phase hybrid approach to solve the complex model, which makes us deeply inspired [36]. So in this study, a multi-adaptive genetic algorithm (MAGA) that combines the characteristics of the two adaptive genetic algorithms was proposed and was compared with these two adaptive genetic algorithms in optimization.

The rest of the research is organized as follows. Section 2 shows the application scenario and the mathematical model. Section 3 is the design of the multi-adaptive genetic algorithm (MAGA). Section 4 demonstrates the experimentation and analysis. Moreover, a conclusion is drawn in section 5.

## Problem descriptions and assumptions

### Facility layout

In the intelligent warehouse of logistics or e-commerce company, taking the unmanned automatic warehouse operation process of Jingdong and Cainiao as the example, after the consumer places an online order, the Automated Storage and Retrieval System (AS/RS) in the warehouse will pick out the goods that the consumer needs according to the order, and transport them to the packaging area by the conveyor belt. Then after the packaging, the package will be transported to the sorting area by the conveyor belt, and then the AGV will perform the sorting operation. The layout of the work scene is shown in Fig 1. The responsibility of the

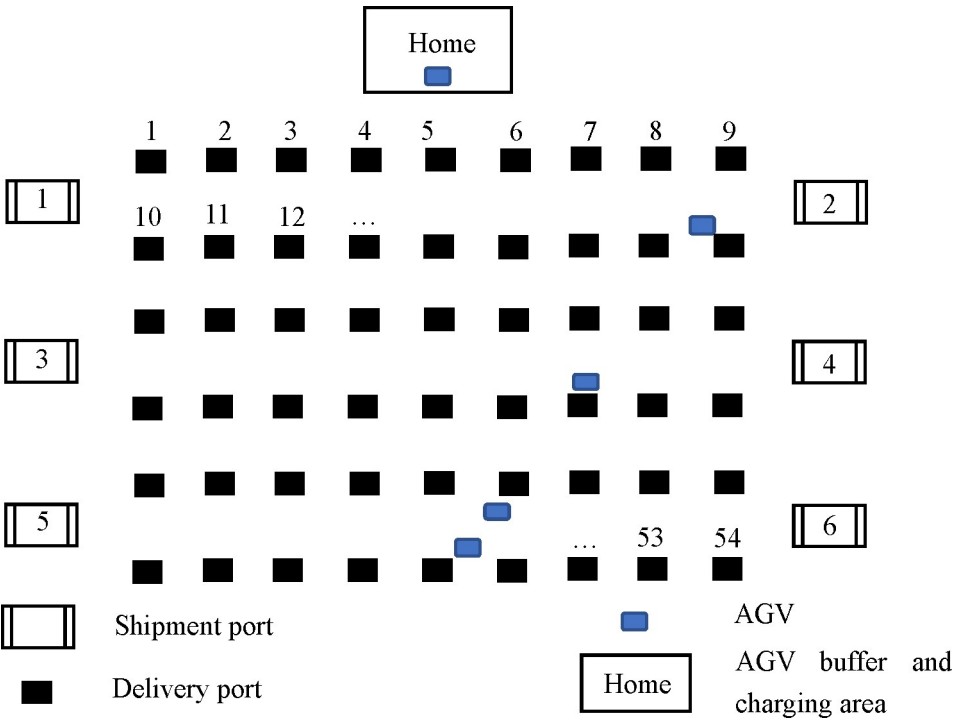

**Fig 1. The layout of the AGV sorting area.**

AGV is to transport the parcels transported by different conveyors (shipment ports) to the corresponding delivery point according to the information of the express waybill in the sorting area. Below the delivery ports are funnels and conveyors that collect the packages and transport them to the distribution area.

## Model derivation

This section develops a mathematical model of AGV scheduling based on three objectives. These three objectives are: (1) minimizing the makespan, (2) minimizing the number of AGVs used, and (3) minimizing the amount of electricity consumed by all AGVs.

The assumptions and limitations for the model development are as follows:

**Assumptions.**

1. The AGVs are parked in the home until scheduling commands are assigned.

2. The travel speed of the AGV can be scheduled, and each order corresponds to an independent AGV speed.

3. The research environment is a free-form grid-like road network, and the road width is sufficient, so there are no traffic problems, collision, deadlock, or conflict.

4. When the AGV sorts the order, it takes the shortest path. The shortest path distance is uniquely determined by the beginning and ending points of the AGV.

5. The loading and unloading time of the AGV is short and negligible.

6. The AGV can stay in the loading/unloading position (shipment port/delivery port).

7. Each AGV only loads one package at a time, that is, each AGV can only perform the sorting work of one order at the same time.

**Notations.**
MS—Makespan
NA—The number of AGVs assigned to the order sorting
E—The amount of electricity consumed by all AGVs
m—The total number of orders
$O_i$—Order number $i$
$i,j$—Index of orders, $i, j$ = 1, 2, 3, ..., m
I—The set of all orders
n—The total number of AGVs in the warehouse
$A^a$—AGV number $a$
$a$—Index of AGVs, $a$ = 1, 2, 3, ..., n
A—The set of all AGVs
$S_i$—The shipment port of $O_i$
$D_i$—The delivery port of $O_i$
H—AGV charging area
$T^a$—$A^a$ assigned to the order sorting
$T_i^a$—$A^a$ assigned to complete the sorting of $O_i$
$vT_i^a$—The speed of $A^a$ when it is assigned to complete the sorting of $O_i$, $v_{min} \leq vT_i^a \leq v_{max}$
$T_{ij}^a$—$A^a$ assigned to complete the sorting of $O_j$ after completing the sorting of $O_i$
$tO_i$—The time when the package in $O_i$ arrive at $S_i$ (i.e., the earliest time when $O_i$ can be sorted)
tST—The time when the order sorting system starts to perform its tasks (the beginning of the whole system)

$t\mathrm{ST}_i$—The time when $O_i$ gets started to be sorted

$t\mathrm{CT}_i$—The time required to complete the sorting of $O_i$ (i.e., the time required for the AGV to take the package from $S_i$ and transport it to $D_i$)

$d\mathrm{CT}_i$—The distance that the AGV needs to complete the sorting of $O_i$ (i.e., the distance from $S_i$ to $D_i$)

$\mathrm{CA}^a$—The current position of $A^a$

$t\mathrm{CA}^a$—The time when $A^a$ is in the current position

$tT_i^a$—The time required for $A^a$ to arrive at the shipment port of $O_i$ from the current position (i.e., the time required for $A^a$ to arrive at $S_i$ from $\mathrm{CA}^a$)

$dT_i^a$—The distance from the current position of $A^a$ to the shipment port of $O_i$ (i.e., the distance from $\mathrm{CA}^a$ to $S_i$)

$tT_{ij}^a$—The time required for $A^a$ to arrive at the shipment port of $O_j$ from the delivery port of $O_i$ (i.e., the time required for $A^a$ to arrive at $S_j$ from $D_i$)

$dT_{ij}^a$—The distance from the shipment port of $O_j$ to the delivery port of $O_i$ (i.e., the distance from $S_j$ to $D_i$)

$\mathrm{EA}_0^a$—The initial state of charge (SOC) of $A^a$'s battery

$\mathrm{E}A^a$—The current state of charge (SOC) of $A^a$'s battery

$\mathrm{ECA}^a$—The charging capacity of $A^a$

$\mathrm{EA}_f^a$—The final state of charge (SOC) of $A^a$'s battery

$\mathrm{EH}T_i^a$—To complete the sorting of $O_i$, it is necessary to perform a charging task first when $\mathrm{E}A^a$ is insufficient

$t\mathrm{EH}T_i^a$—The time required for $A^a$ to perform the charging task ($\mathrm{EH}T_i^a$)

$t\mathrm{E}T_i^a$—The charging time for $A^a$ in the charging task ($\mathrm{EH}T_i^a$)

$t\mathrm{H}T_i^a$—The round-trip time for $A^a$ to travel between the work area ($\mathrm{CA}^a$) and the charging area (H)

$d\mathrm{H}T_i^a$—The distance that $A^a$ travels between the work area ($\mathrm{CA}^a$) and the charging area (H)

$\mathrm{EH}T_i^a$—The amount of electricity required for $A^a$ to complete the sorting of $O_i$ from the current position and return to the charging area (H)

$\mu_i$—The ratio of power consumption to time when AGV's speed is $vT_i^a$ (i.e., $\mu_i$ is the amount of power consumed per unit time at speed $vT_i^a$), since $v_{min} \leq vT_i^a \leq v_{max}$, $\mu_{min} \leq \mu_i \leq \mu_{max}$

$ET_{iH}^a$—The minimum power consumption required for $A^a$ to move from the delivery port of $O_i$ ($D_i$) to the charging area (H)

**Variables—Sorting assignment.** Use the 0–1 variable $T_i^a$ as the index for order assignment:

$$T_i^a = \begin{cases} 1, & \textit{if } A^a \textit{ is assigned to sort } O_i \\ 0, & \textit{otherwise} \end{cases} \qquad (1)$$

Use the 0–1 variable $T^a$ to help calculate the number of AGVs used in sorting systems. As long as AGV-$A^a$ participates in order sorting, $T^a$ equals 1:

$$T^a = \begin{cases} 1, & \textit{if } \exists i \in I, T_i^a = 1 \\ 0, & \textit{otherwise} \end{cases} \qquad (2)$$

Use the 0–1 variable $T_{ij}^a$ as the index for order sequencing:

$$T_{ij}^a = \begin{cases} 1, & \textit{if } A^a \textit{ is assigned to sort } O_j \textit{ after } \textit{sorting } O_i \\ 0, & \textit{otherwise} \end{cases} \qquad (3)$$

**Variables—Speed.** The time AGV-A$^a$ needed to pick up packages, the time AGV-A$^a$ needed to to sort orders and its charging time are determined by its adjustable speed:

$$tCT_i = dCT_i/vT_i^a \tag{4}$$

$$tT_i^a = dT_i^a/vT_i^a \tag{5}$$

$$tHT_i^a = dHT_i^a/v_{min} \tag{6}$$

$$tT_{ij}^a = dT_{ij}^a/vT_j^a \tag{7}$$

**Objective function.** Makespan is the time when the last order is sorted minus the start time of the system:

$$MS = max\{(tST_i + tCT_i)\} - tST \tag{8}$$

Eq 9 is used to calculate the number of AGVs involved in sorting:

$$NA = \sum_{a=1}^{n} T^a \tag{9}$$

The power consumption is calculated by the initial values, charge values and residual value of AGVs' batteries. It can also be calculated by the sorting order time and the power consumption coefficient:

$$E = \sum_{a=1}^{n}(EA_0^a + ECA^a - EA_f^a) = \sum_{i=1}^{m}\sum_{a=1}^{n}\{\mu_{min}tHT_i^a + \mu_i(tT_i^a + tCT_i)\} \tag{10}$$

Subject to:

$$tST_i \geq tO_i, \forall i \in I \tag{11}$$

$$tST_i \geq T_i^a(tCA^a + tEHT_i^a + tT_i^a), \forall i \in I, a \in A \tag{12}$$

$$tST_j \geq T_{ij}^a(tST_i + tCT_i + tT_{ij}^a), \forall i,j \in I, i \neq j, a \in A \tag{13}$$

$$tEHT_i^a = \begin{cases} 0, & if \ EA^a \geq EHT_i^a \\ tET_i^a + tHT_i^a, & if \ EA^a < EHT_i^a \end{cases} \tag{14}$$

$$EHT_i^a = \mu_i(tT_i^a + tCT_i) + ET_{iH}^a \tag{15}$$

$$\sum_a T_i^a = 1, \forall i \in I \tag{16}$$

$$T_{ij}^a + T_{ji}^a \leq 1, \forall i,j \in I, i \neq j, a \in A \tag{17}$$

Where Eqs 11, 12 and 13 are time constraints, respectively ensuring that the start time of the order-sorting is later than the arrival time of the order, the arrival time of the AGV, and the completion time of the previous order. Eq 14 is the time of the charging task, which is the power constraint of the system. If the power of the AGV is insufficient to complete the order and return to the charging area, the AGV will need to perform the charging task. Eq 15 calculates the amount of electricity required for the AGV to complete the sorting of $O_i$ and return to the charging area. Eq 16 determines that each order can and can only be executed once by

one AGV. Eq 17 makes sure that the responsible order sequence is deterministic for each AGV.

## Multi-objective evaluation

Pareto efficiency implies that resources are allocated in the most efficient manner [37]. The overall fitness function is expressed as follows

$$f(x) = \sum \theta_\alpha \omega_\alpha f_\alpha(x) \tag{18}$$

Where $\omega_\alpha$ is the weight of the $\alpha_{th}$ objective function ($\Sigma \omega_\alpha = 1$), and $\theta_\alpha$ is the coefficient of the $\alpha_{th}$ objective function for obtaining a range of approximate values among the objectives.

In this study, the weights of the three targets were 0.6, 0.24, and 0.16, respectively. To have similar ranges of variation for the three objectives, the adjustment coefficients were 1, 30, and 20, respectively. The following formula calculates the overall fitness function.

$$f(x) = 0.6(MS) + 7.2(NA) + 3.2(E) \tag{19}$$

## Algorithm design

In recent years, there have been many improvements in adaptive genetic algorithms. The two most commonly used ones are:

1. Adaptive adjustment of crossover and mutation rate based on individual fitness values to reduce damage to good genes (AGA1).

2. Adding population entropy to the algorithm, adaptively adjusting the crossover and mutation rate based on the population entropy to maintain the diversity of the population and improve the global search ability of the algorithm (AGA2).

It is not enough to consider the diversity of the population. The variety of individual genes also has a substantial impact. Therefore, it is necessary to add the operation of improving the genetic diversity to the algorithm. Besides, the crossover and mutation rates can be adaptively adjusted in combination with the population entropy and the individual fitness value to ensure that the population is diverse and those competent individuals have a higher probability of entering the next generation.

In this research, we combined the characteristics of these two adaptive genetic algorithms and improved them. A multi-adaptive genetic algorithm (MAGA) is proposed as follows.

## Chromosome representation and encoding

The number of genes in the chromosome is twice the number of orders. The first half of the chromosome indicates the assignment of the order sorting, and the second half indicates the speed at which the AGV sorts the order. For example, if the order number is 30, the number of genes on the chromosome will be 60. The chromosome is shown in Fig 2, which indicates that order No. 1 is sorted by AGV No. 9 at a speed of 1.0m/s, order No. 2 is sorted by AGV No. 8 at a speed of 1.2m/s, and order No. 30 is sorted by AGV No. 3 at a speed of 1.5 m/s.

## Fitness function

The total fitness value is expressed by F, and it will be calculated based on Eq 20.

$$F = 1/f(x) \tag{20}$$

| 9 | 8 | ... | 3 | 1.0 | 1.2 | ... | 1.5 |

Gene position    1    2    ...    30    31    32    ...    60

**Fig 2. A random chromosome.**

## Initializing population

The initial population is randomly generated, but it must be ensured that the initial population has the appropriate diversity, that is the population entropy cannot be too small.

**The representation and calculation of population entropy.** The solution space is divided into M non-overlapping regions ($Q_1$, $Q_2$, . . ., $Q_M$). $p_i$ ($i$ = 1, 2, . . ., M) is used to indicate the probability that an individual in the population belongs to $Q_i$. The population entropy of the $t_{th}$ generation ($S(t)$) can be expressed as:

$$S(t) = -\sum_{i=1}^{M} p_i \ln p_i \tag{21}$$

In order to estimate the population entropy, the range $[(1-\alpha)F_{min}, (1+\alpha)F_{max}]$ is used instead of the solution space, where $F_{min}$ is the minimum value of the fitness value from the initial iteration to the $t_{th}$ generation, $F_{max}$ is the maximum one, and $\alpha$ ($0 < \alpha < 0.1$) is an expansion coefficient. Then divide the entire interval into N (N is the number of individuals in the population) regions. Use $l_i$ ($i$ = 1, 2, . . ., N) to denote the number of individuals whose fitness value belongs to the $i_{th}$ region. The estimated value of $p_i$ is calculated using

$$\widehat{p_i} = l_i/N \tag{22}$$

Substituting the estimated value of $p_i$ ($\widehat{p_i}$) into Eq 21 yields the estimate of the population entropy ($\widehat{S(t)}$):

$$\widehat{S(t)} = -\sum_{i=1}^{N} \widehat{p_i} \ln \widehat{p_i} \tag{23}$$

## Selection and elitism

In this study, we use the Roulette Wheel Selection [38] to select. The best individual in each generation is transferred directly to the next generation in the elitism step.

## Crossover

This study employs a multi-point crossover. Since the first half of the chromosome represents the sorting assignment, and the second half represents the speeds of AGVs, there is a correlation between the two. In order to ensure that the proper individual is not destroyed by crossover operation, after the first half of its chromosome exchanges fragments (sequence of genes) with the other chromosome, the corresponding part of the latter half has to exchange fragments. For example, if the order number is 30, the number of genes on the chromosome will be 60. When genes between the $2_{nd}$ and $30_{th}$ on chromosomes are crossed over, genes between the $32_{nd}$ and the $60_{th}$ need to be correspondingly crossed over, as shown in Fig 3.

The crossover rate ($p_c$) of this study is multi-adaptively adjusted. Firstly, the basic number of crossover rate of the $t_{th}$ generation ($p_{c1}$) is determined according to the population entropy by using

$$\beta = S(t)/S_{max} \tag{24}$$

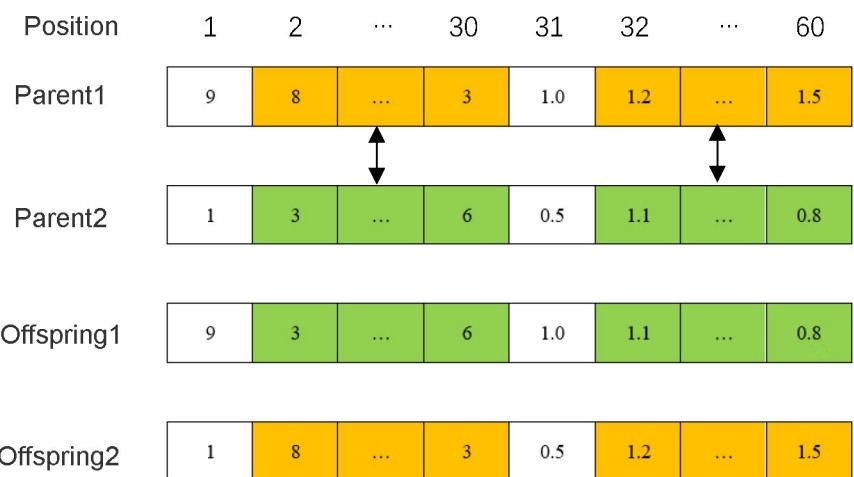

**Fig 3. An example of crossover.**

$$p_{c1} = p_{c2} + p_{c3}(1 - \beta) \tag{25}$$

Where $S_{max}$ is the maximum possible value of the population entropy, that is, $S_{max} = \ln N$. $p_{c2}$ and $p_{c3}$ are parameters that can be adjusted. When the population diversity becomes smaller, the basic number of the crossover rate becomes larger to increase the diversity of the population.

Secondly, the crossover rate of an individual ($p_c$) is determined according to the fitness value of the individual by Eq 26.

$$p_c = p_{c1} F_{max}/(\gamma F) \tag{26}$$

Where F is the larger fitness value of the two crossing-over individuals. If the individual fitness value is large, the crossover rate will be low, so that the structure of a good individual can be destroyed as little as possible. $\gamma$ is an adjustment coefficient to ensure that individuals with small current fitness values are also likely to enter the next generation.

## Mutation

In order to maintain the diversity of the population and individuals, the 0–1 variable ($X^k_{ij}$) was added to help. If the individual $i$ is different from the individual $j$ in the $k_{th}$ gene, then $X^k_{ij} = 1$ otherwise $X^k_{ij} = 1$. The degree of diversity of the $k_{th}$ gene of all individuals of the $t_{th}$ generation population ($Y^k_t$) can be expressed as

$$Y^k_t = \sum_{i=1}^{N-1} \sum_{j=i+1}^{N} X^k_{ij} \tag{27}$$

In the mutation operation, the position of the mutation is determined according to the value of $Y^k_t$—

the smaller the genetic diversity, the greater the probability that the gene position is selected to perform the mutation operation (based on the Roulette Wheel Selection).

The mutation rate ($p_m$) of this study is also multi-adaptively adjusted. Firstly, the basic number of the mutation rate of the $t_{th}$ generation ($p_{m1}$) is calculated according to the population entropy by Eq 28.

$$p_{m1} = p_{m2} + p_{m3}(1 - \beta) \tag{28}$$

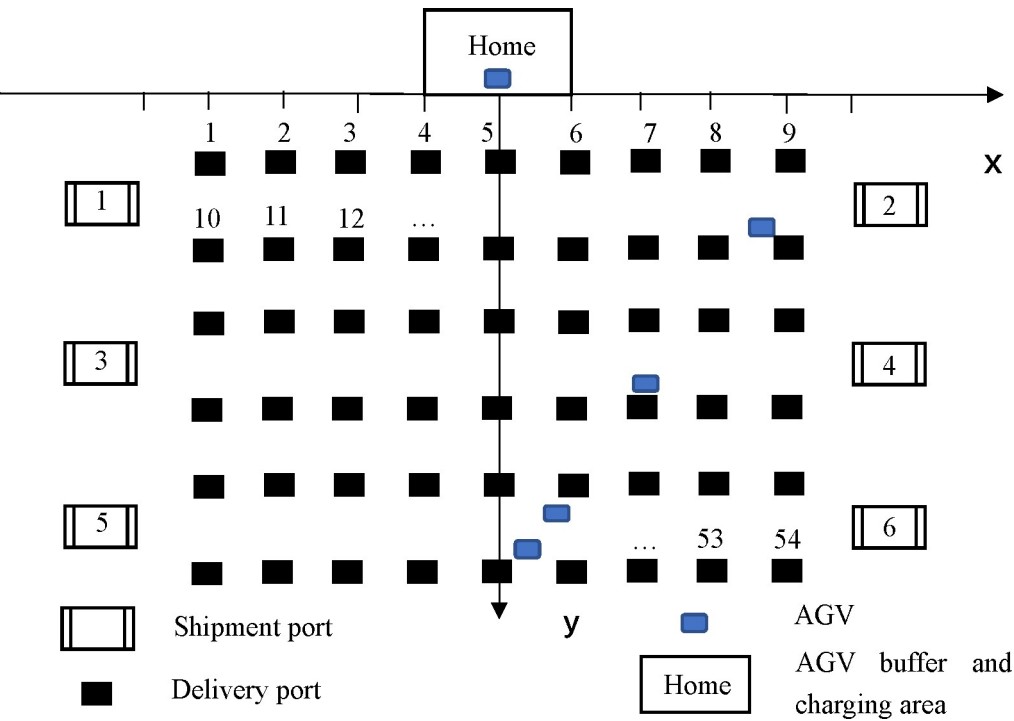

**Fig 4. The upper half of the warehouse coordinate system.**

Where $p_{m2}$ and $p_{m3}$ are parameters that can be adjusted. If the population diversity becomes small, $p_{m1}$ will become large, which is conducive to the generation of new individuals and increase the diversity of the population.

Then the mutation rate of an individual ($p_m$) is determined according to the individual fitness value using

$$p_m = p_{m1} F_{max} / (\gamma F) \tag{29}$$

Where F is the fitness value of the mutated individual. If the individual fitness value is getting larger, the mutation rate will be lower, so that the gene of the excellent individual can be protected.

The characteristics of the Multi-Adaptive Genetic Algorithm (MAGA) proposed in this study are summarized as follows.

1. The crossover rate and mutation rate are multiple adaptively adjusted.

2. The site of the mutation is not random but varies according to the degree of the individual genetic diversity.

3. The initial population is to meet certain conditions, which is conducive to finding a better solution faster.

Thus MAGA is an algorithm with an excellent ability of global optimization, and it can make good results on the problem of a time-cost optimization solution for its convergence rate.

## Computational results and discussion

To validate the model, two-scale numerical experiments have been conducted. In the first experiment, there were 30 orders ($O_1$, $O_2$, . . ., $O_{30}$) needed to be sorted, 10 AGVs ($A^1$, $A^2$, . . ., $A^{10}$) in the warehouse, and the AGV speed varied from 0.5m/s to 1.5m/s. While in the second experiment, there were 50 orders ($O_1$, $O_2$, . . ., $O_{50}$), 20 AGVs ($A^1$, $A^2$, . . ., $A^{20}$) and the AGV speed adjustable range was also 0.5m/s-1.5m/s.

Taking the Home as the coordinate origin, a coordinate system is established to determine the coordinates of the shipment port and the delivery port, shown in Fig 4, and then the moving distance of AGVs (each unit on the x-axis or y-axis stands for 2 meters) is calculated. For example, the coordinates of the Home are (0, 0), the coordinates of the No. 3 shipping port are (-10, 7), the coordinates of the No. 6 shipping port are (10, 11), the coordinates of the No. 2 delivery port are (-6, 2), the coordinates of the No. 10 delivery port are (-8, 2), and the coordinates of the No. 53 delivery port are (6, 12) (Fig 3 is only a schematic diagram of half of the warehouse, so there are 12 shipment ports and 108 delivery ports). An example of the data content of the order is shown in Table 1.

The superiority of the proposed algorithm was verified by comparing the following algorithms:

AGA1: Adaptive adjustment of crossover and mutation rates based on individual fitness values, $p_c = p_{c1}F_{max}/(\gamma F)$, $p_m = p_{m1}F_{max}/(\gamma F)$.

AGA2: Adaptive adjustment of crossover and mutation rates based on the population entropy, $p_c = p_{c1}+p_{c2}(1-\beta)$, $p_m = p_{m1}+p_{m2}(1-\beta)$.

MAGA: The hybrid improvement algorithm proposed in Section 3 of this study, $p_c = p_{c1}F_{max}/(\gamma F)$, $p_{c1} = p_{c2}+p_{c3}(1-\beta)$, $p_m = p_{m1}F_{max}/(\gamma F)$, $p_{m1} = p_{m2}+p_{m3}(1-\beta)$.

In order to enhance the representativeness of the experiments, a control with sufficient AGV power and insufficient AGV power was set in both experiments. Based on the experimental approach, the best settings are as follows:

Experiment 1

AGA1: $p_{c1} = 0.6$, $p_{m1} = 0.04$, $\gamma = 2$.

AGA2: $p_{c1} = 0.6$, $p_{c2} = 0.3$, $p_{m1} = 0.04$, $p_{m2} = 0.06$, $\alpha = 0.05$.

MAGA: $p_{c2} = 0.6$, $p_{c3} = 0.3$, $p_{m2} = 0.04$, $p_{m3} = 0.06$, $\gamma = 2$, $\alpha = 0.05$.

The algorithms were run 10 times, with each run of a population size of 60 in 350 iterations and their results are shown in Table 2.

Experiment 2

AGA1: $p_{c1} = 0.7$, $p_{m1} = 0.07$, $\gamma = 2$.

AGA2: $p_{c1} = 0.7$, $p_{c2} = 0.2$, $p_{m1} = 0.07$, $p_{m2} = 0.03$, $\alpha = 0.08$.

MAGA: $p_{c2} = 0.7$, $p_{c3} = 0.2$, $p_{m2} = 0.07$, $p_{m3} = 0.03$, $\gamma = 2$, $\alpha = 0.08$.

The algorithms were run 10 times, with each run of a population size of 100 in 500 iterations, and their results are shown in Table 3.

The performances of the three algorithms in the experiments are shown in Figs 5–8. The changes in population entropy are shown in Fig 9.

It can be seen from Figs 5–9 that the AGA1 performs well in the convergence rate because the crossover rate and mutation rate of AGA1 are adaptively adjusted based on the individual

**Table 1. Table of order information.**

| Order ID | Shipment port No. | Delivery port No. | Order arrival time ($tO_i$) |
|---|---|---|---|
| 1 | 12 | 29 | 2019-04-10-13:10:00 |
| 2 | 5 | 10 | 2019-04-10-13:10:00 |
| 3 | 3 | 67 | 2019-04-10-13:10:01 |

**Table 2. Test results of optimization algorithms for Experiment 1.**

| Algorithms | | Experiment 1-1(fully charged) | | | Experiment 1-2(low battery) | | |
| --- | --- | --- | --- | --- | --- | --- | --- |
| | | The best result | Mean result | Mean CPU time | The best result | Mean result | Mean CPU time |
| AGA1 | $f(x)$ | 185.9 | 189.4 | 409.6 Sec | 203.9 | 207.2 | 397.3 Sec |
| | MS | 146.2 | 139.6 | | 171.5 | 168.2 | |
| | NA | 8 | 8.8 | | 8 | 8.6 | |
| | E | 12.69 | 13.21 | | 13.57 | 13.85 | |
| AGA2 | $f(x)$ | 183.1 | 186.6 | 411.8 Sec | 195.9 | 201.9 | 410.6 Sec |
| | MS | 119.5 | 130.6 | | 163.2 | 168.2 | |
| | NA | 10 | 9.3 | | 8 | 8.5 | |
| | E | 12.32 | 12.91 | | 12.62 | 12.44 | |
| MAGA | $f(x)$ | 171.9 | 176.3 | 415.2 Sec | 182.6 | 187.5 | 412.5 Sec |
| | MS | 118.6 | 124.4 | | 146.1 | 158.7 | |
| | NA | 9 | 8.9 | | 8 | 8.2 | |
| | E | 11.22 | 11.73 | | 11.67 | 10.39 | |

fitness value. If the individual fitness value is large, the rates of crossover and mutation will be low, so that the great gene sequence is more natural to enter the next generation without being destructed. Therefore, the application of AGA1 can help find a suitable solution quickly. However, the shortcomings of AGA1 is also apparent, that is, the global search ability is not strong so that it is easy to run into the local optimization solution. The mean value of $f(x)$ and the value of population entropy shown in these figures can be well supported by this point. The population entropy of AGA1 decreases rapidly with the increase of iteration numbers, and the mean value of population and optimal value are very close after 150 iterations, which indicates that individuals in the population have converged and it is challenging to produce better individuals.

The performance of AGA2 is precisely the opposite of AGA1. AGA2 has a stronger global search ability but a slower convergence rate. Since the crossover rate and mutation rate of AGA2 are adaptively adjusted based on the value of the population entropy, if the population entropy becomes small, the crossover rate and the mutation rate will increase, thereby

**Table 3. Test results of optimization algorithms for Experiment 2.**

| Algorithms | | Experiment 2-1(fully charged) | | | Experiment 2-2(low battery) | | |
| --- | --- | --- | --- | --- | --- | --- | --- |
| | | The best result | Mean result | Mean CPU time | The best result | Mean result | Mean CPU time |
| AGA1 | $f(x)$ | 274.0 | 278.2 | 1609.6 Sec | 319.6 | 325.3 | 1604.1 Sec |
| | MS | 131.5 | 147.9 | | 192.6 | 189.3 | |
| | NA | 18 | 17.1 | | 17 | 17.5 | |
| | E | 20.48 | 20.74 | | 25.5 | 26.8 | |
| AGA2 | $f(x)$ | 271.5 | 274.5 | 1611.7 Sec | 294.4 | 301.3 | 1599.7 Sec |
| | MS | 149.8 | 168.6 | | 156.2 | 176.5 | |
| | NA | 16 | 14.7 | | 18 | 16.9 | |
| | E | 20.74 | 21.10 | | 22.21 | 23.03 | |
| MAGA | $f(x)$ | 256.5 | 258.3 | 1620.3 Sec | 274.2 | 279.1 | 1634.8 Sec |
| | MS | 133.3 | 136.7 | | 184.9 | 182.1 | |
| | NA | 16 | 15.9 | | 13 | 13.5 | |
| | E | 19.15 | 19.30 | | 21.78 | 22.70 | |

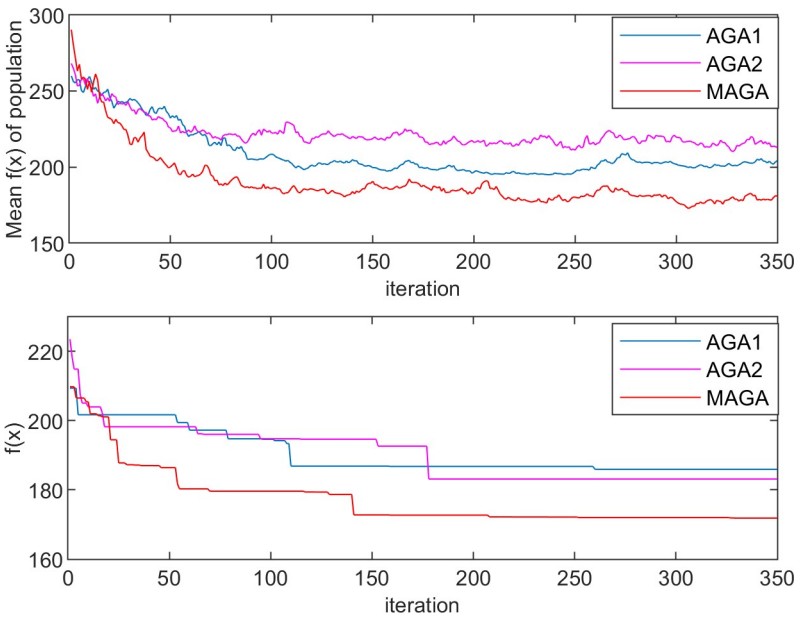

**Fig 5. Performances of the different algorithms for Experiment 1–1.**

generating more new individuals to increase the diversity of the population. It can be seen from Fig 9 that the population entropy of AGA2 has been maintained at a relatively high level. So, if AGA2 with a more diverse population is applied, there will be a higher probability of finding a better solution. However, the high population entropy of AGA2 means that individuals are scattered, so it is difficult for the good genes of different gene positions to converge on the same individual. Moreover, the crossover rate and the mutation rate are only related to the current population entropy, which results in the difficulty to retain the high fitness value of the individual genetic structure. Therefore, the convergence speed of AGA2 is slow.

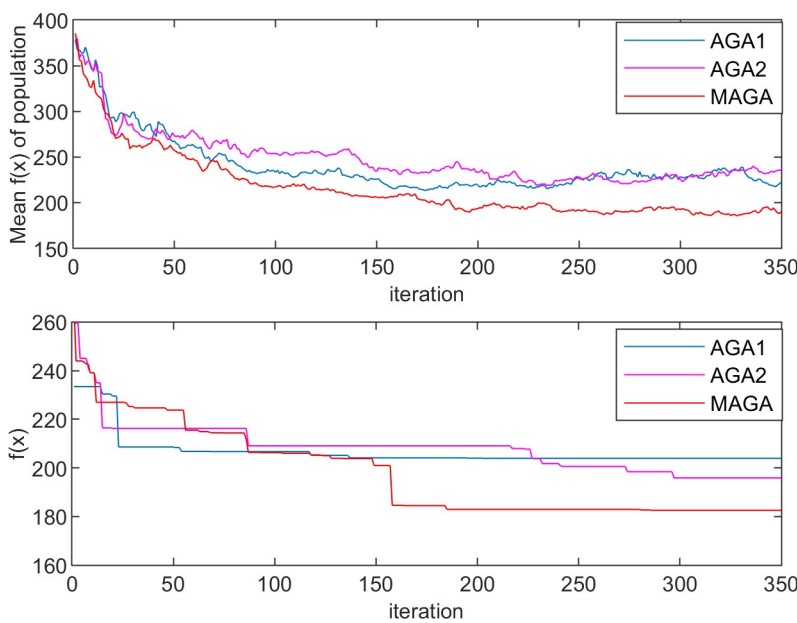

**Fig 6. Performances of the different algorithms for Experiment 1–2.**

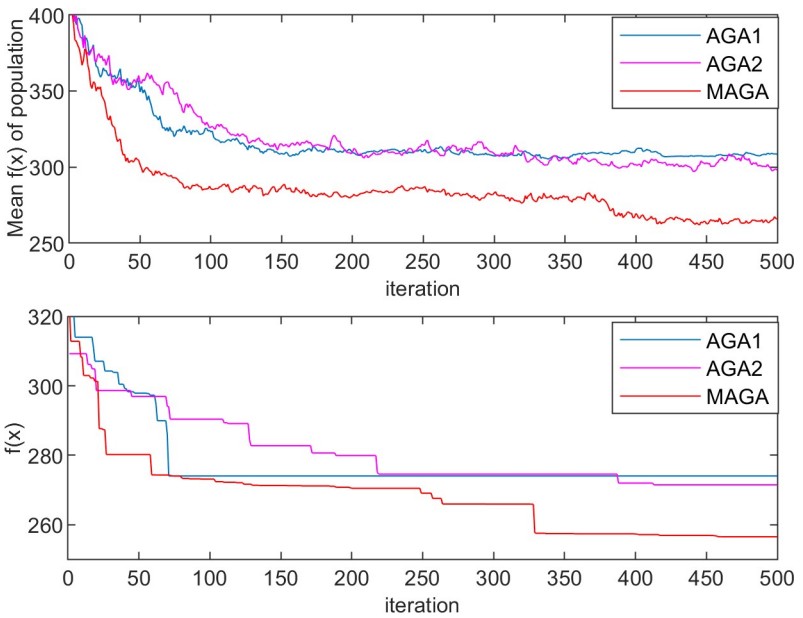

**Fig 7. Performances of the different algorithms for Experiment 2–1.**

Among the three algorithms, the MAGA algorithm performs best. It adaptively adjusts the cardinality of population crossover rate and mutation rate, and selects the gene position with low genetic diversity to perform mutation operation with higher probability, so that the population entropy is maintained at the appropriate level. What is more, the individual crossover rate and mutation rate can be further changed with the change of fitness value to protect the current excellent gene and good gene structure, and thus guarantee the retaining of MAGA's good global search ability and convergence rate.

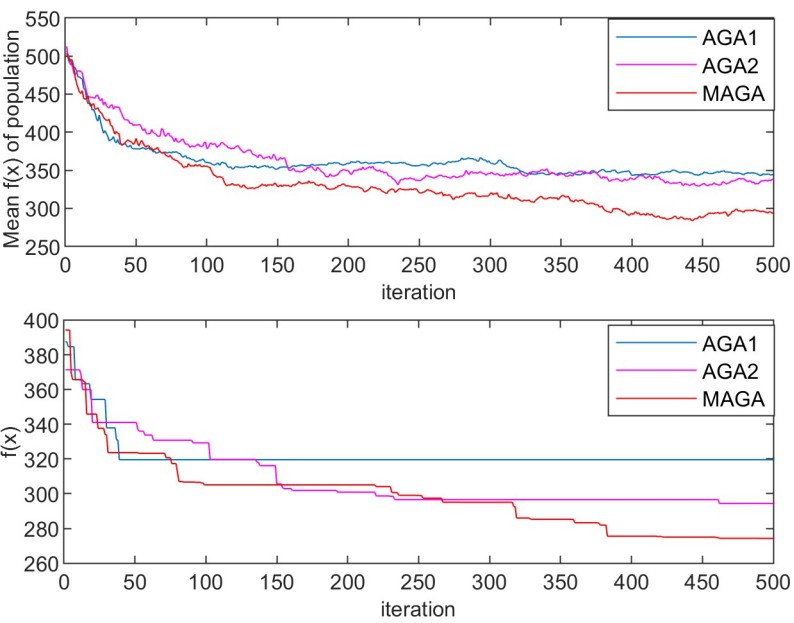

**Fig 8. Performances of the different algorithms for Experiment 2–2.**

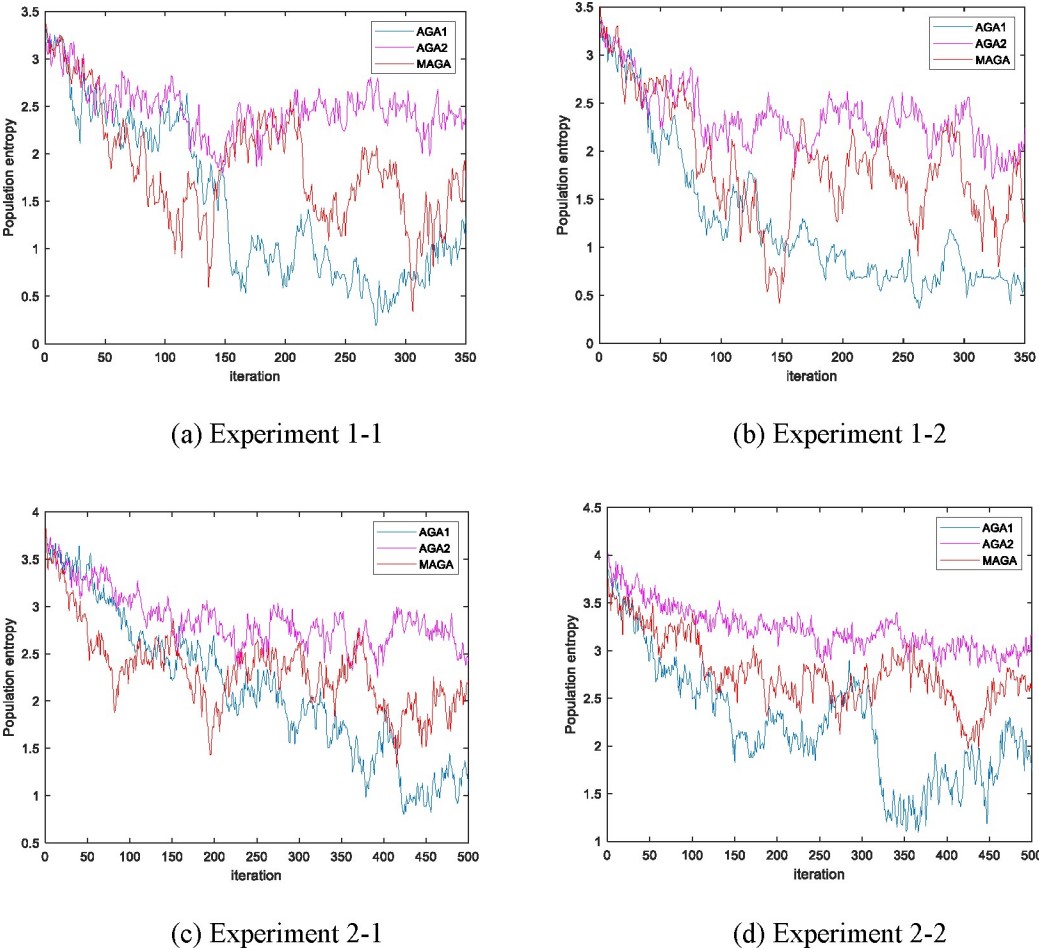

(a) Experiment 1-1 (b) Experiment 1-2

(c) Experiment 2-1 (d) Experiment 2-2

**Fig 9. Population entropy of the different algorithms for all Experiments.**

In several experiments, the optimal distribution results obtained by MAGA are shown in Figs 11–14. The scheduling results are shown in Gantt charts, and the legend is shown in Fig 10.

The effectiveness of MAGA is verified by comparing data in the AGV scheduling system before and after the optimization (shown in Table 4). Taking Experiment 2–2 as an example, Fig 15 shows the changes of various data in the system when using MAGA to optimize.

It can be seen that the model and algorithms proposed in this study reduced the makespan while reducing the number of AGVs, indicating that the operational efficiency of the AGV is significantly improved. Besides, the power consumption was reduced while the makespan was reduced, indicating that the mathematical model arranged reasonable AGVs to sort the orders at reasonable speeds.

In summary, the mathematical model and optimization algorithms, especially the MAGA, have achieved success in reducing makespan, reducing the number of AGVs used, and reducing the total power consumption in AGV scheduling. What is more, the average optimization range is around 30%, which verifies the effectiveness of the model and the algorithms.

## Conclusion

This research focused on the multi-objective AGV scheduling in an automatic sorting system using two different adaptive genetic algorithms (AGA) and one multi-adaptive genetic

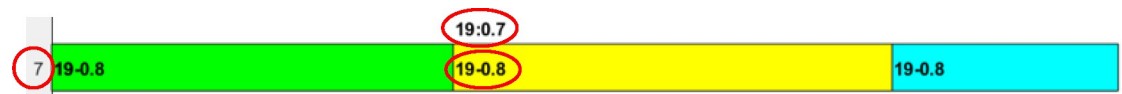

Order 19 is executed by AGV No. 7 at a speed of 0.8 m/s, and the power consumption is 0.7.

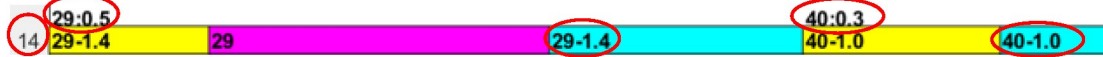

Order 29 is executed by AGV No. 14 at a speed of 1.4 m/s, and the power consumption is 0.5.
Order 40 is executed by AGV No. 14 at a speed of 1.0 m/s, and the power consumption is 0.3.

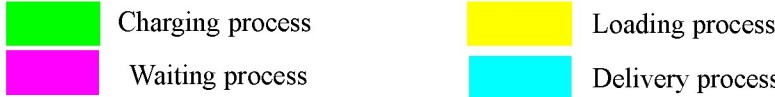

Charging process: The remaining battery level of AGV-A$^a$ is not enough to complete the order $O_i$ and return to the Home area, which needs to be recharged before taking order task.

Waiting process: AGV-A$^a$ arrives at the shipment port before the goods in the order $O_i$, so the AGV needs to wait for the goods to arrive.

Loading process: The process of AGV-A$^a$ going from the current location to the shipment port of the order $O_i$.

Delivery process: The process of AGV-A$^a$ going from the shipment port to delivery port of the order $O_i$.

**Fig 10. The legend of the Gantt chart.**

algorithm (MAGA). Taking into account of changes in AGV battery power and AGV speed, combined with order assignment, charging task assignment and speed determination, a multi-objective mathematical model was developed to minimize the makespan, number of AGVs

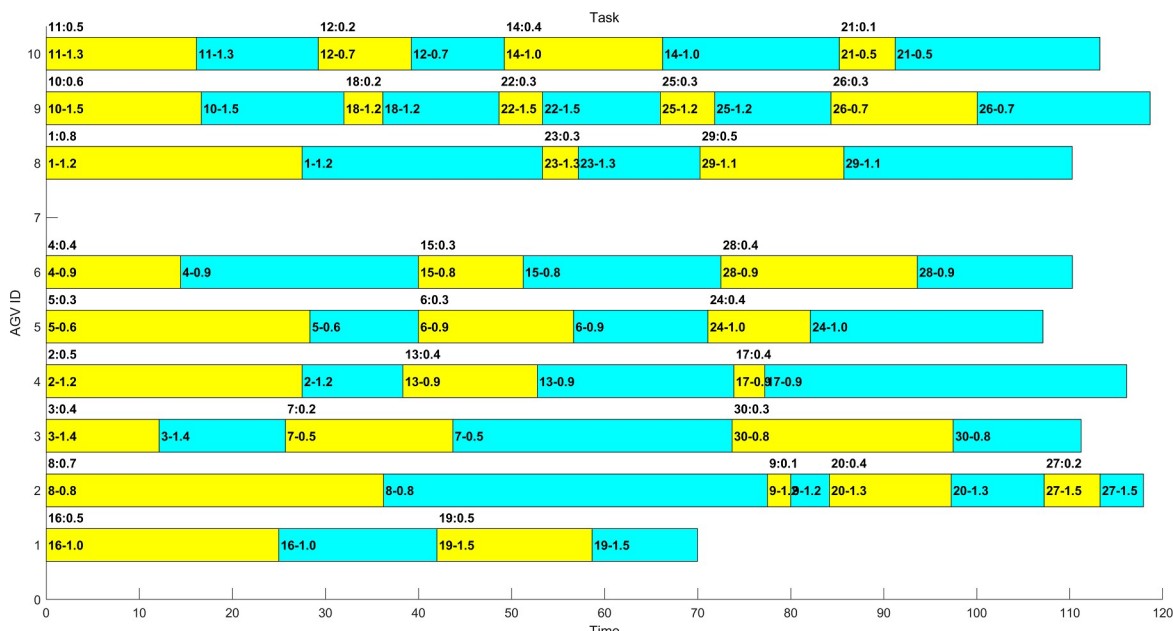

**Fig 11. Gantt chart of the schedule of the example 1–1 after optimization by MAGA.**

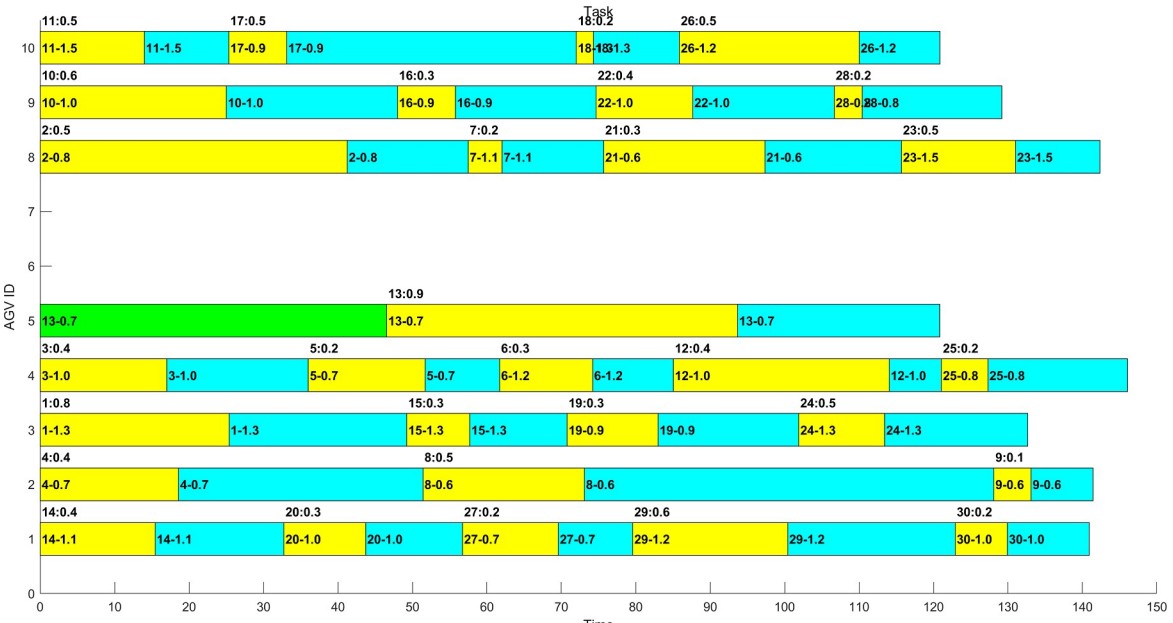

**Fig 12. Gantt chart of the schedule of the example 1–2 after optimization by MAGA.**

used, and amount of electricity consumed by all AGVs to guide AGV scheduling in logistics sorting operations. Comparative numerical experiments were carried out, and the near-optimum schedules of the multi-objective function were successfully obtained. These schedules make it clear with regard to which order is sorted by which specific AGV at what speed. The comparison of the results of the three algorithms showed that the MAGA is superior to the other two adaptive genetic algorithms. A careful comparison of the data before and after

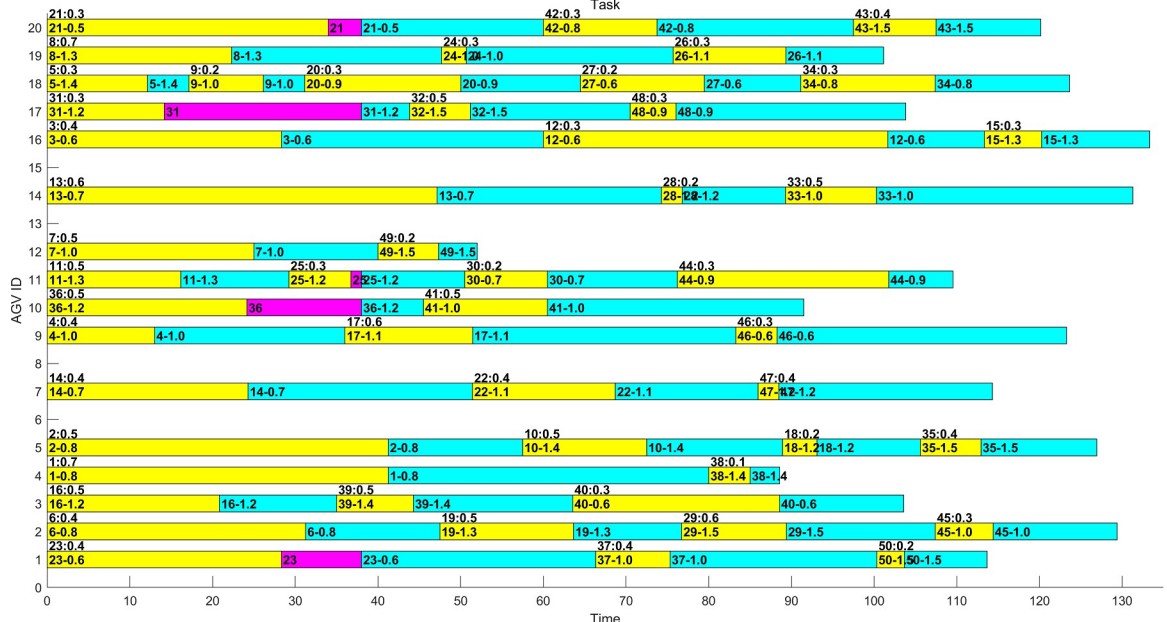

**Fig 13. Gantt chart of the schedule of the example 2–1 after optimization by MAGA.**

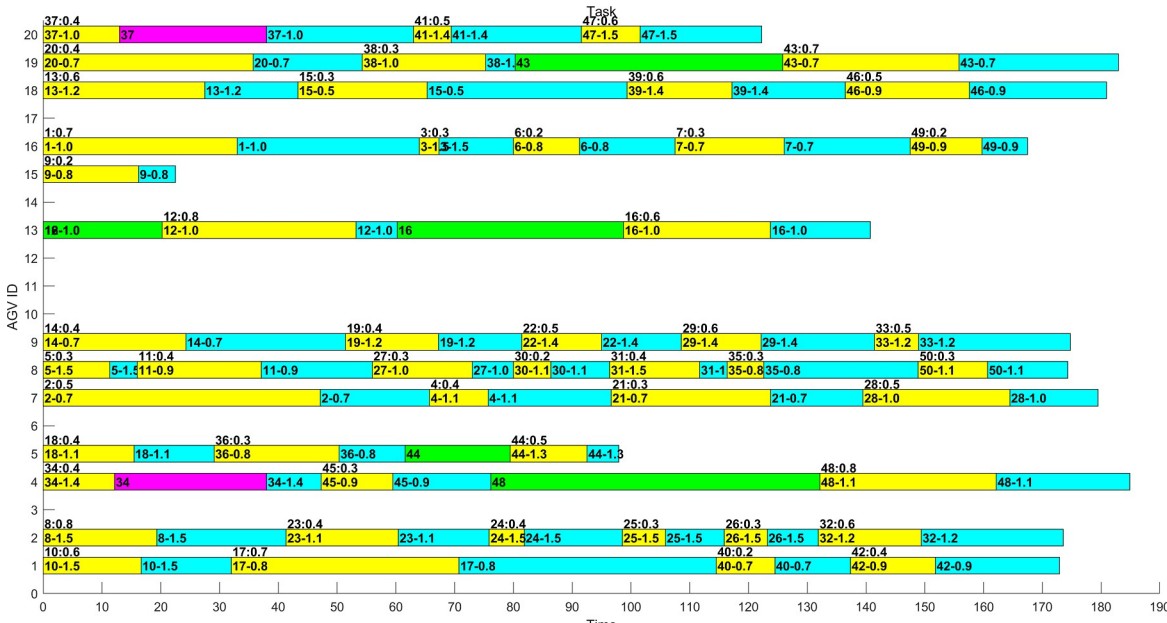

**Fig 14. Gantt chart of the schedule of the example 2–2 after optimization by MAGA.**

optimization revealed that the sorting of all orders could be completed at suitable speeds by fewer AGVs, which results in a shorter makespan, a smaller number of AGVs used, and less total power consumption. Moreover, the value of each objective optimized was reduced by about 30%, which emphatically proved the effectiveness of the model and MAGA.

The major contributions of this paper are as follows:

- The work discussed in this paper can effectively improve the operation efficiency of AGV in enterprise intelligent warehouses.

**Table 4. Data comparison before and after optimization by MAGA.**

| Experiment | | Mean value of 60 or 100 scheduling schemes before optimization | After optimization | Percentage of optimization |
|---|---|---|---|---|
| 1–1 | $f(x)$ | 290.3 | 171.9 | 40.79% |
| | MS | 297.9 | 118.6 | 60.19% |
| | NA | 9.6 | 9 | 6.25% |
| | E | 13.31 | 11.22 | 15.70% |
| 1–2 | $f(x)$ | 385.5 | 182.6 | 52.64% |
| | MS | 443.3 | 146.1 | 67.04% |
| | NA | 9.4 | 8 | 14.89% |
| | E | 16.13 | 11.67 | 27.65% |
| 2–1 | $f(x)$ | 412.9 | 256.5 | 37.88% |
| | MS | 344.2 | 133.3 | 61.27% |
| | NA | 18.3 | 16 | 12.57% |
| | E | 23.24 | 19.15 | 17.60% |
| 2–2 | $f(x)$ | 503.8 | 274.2 | 45.57% |
| | MS | 460.8 | 184.9 | 59.87% |
| | NA | 18.4 | 13 | 29.35% |
| | E | 29.69 | 21.78 | 26.64% |

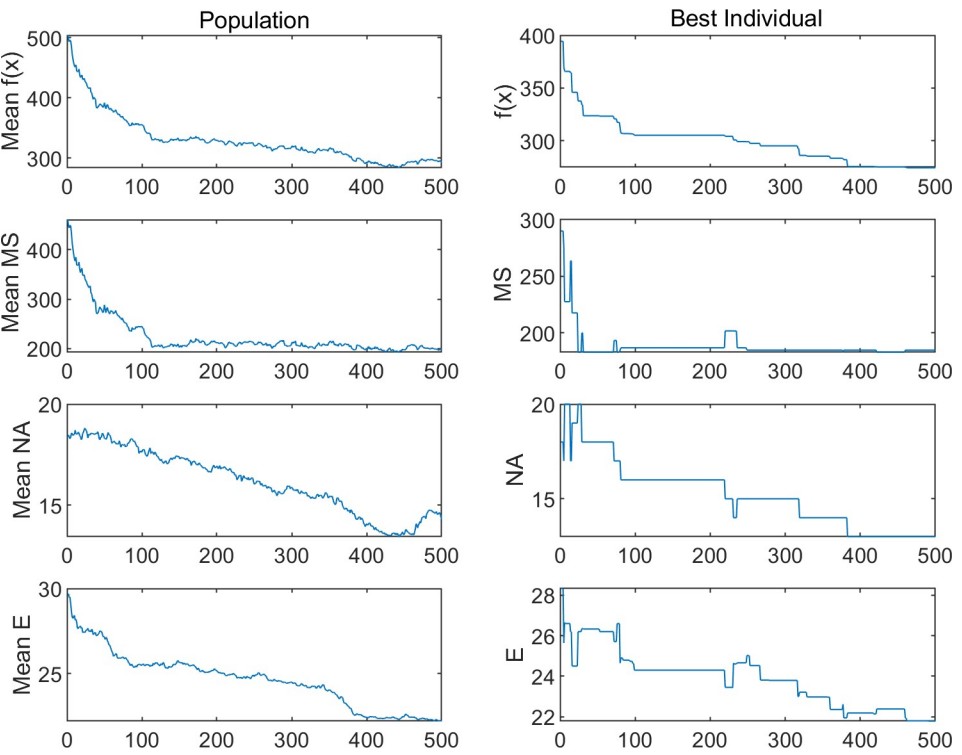

**Fig 15. Data changes in the system for Experiment 2–2 optimization by MAGA.**

- The multi-adaptive genetic algorithm proposed in this paper has strong global search ability and fast optimization speed, which may help other scholars get the approximate optimal solution faster and better under the same conditions.

- The scheduling model developed in this paper may inspire AGV scheduling in different research fields, such as FMS, to take speed as one of the variables and energy consumption as one of the objectives.

## Supporting information

**S1 Appendix. Programming codes for MAGA.**
(PDF)

**S1 Data. Order data.**
(XLSX)

## Acknowledgments

This research was supported by the National Key R&D Program of China (No. 2018YFB1601603).

## Author Contributions

**Conceptualization:** Yubang Liu, Shouwen Ji, Dong Guo.

**Data curation:** Yubang Liu, Zengrong Su, Dong Guo.

**Formal analysis:** Yubang Liu, Shouwen Ji.

**Funding acquisition:** Shouwen Ji.

**Investigation:** Yubang Liu, Shouwen Ji, Zengrong Su, Dong Guo.

**Methodology:** Yubang Liu, Shouwen Ji.

**Project administration:** Yubang Liu.

**Resources:** Yubang Liu, Shouwen Ji, Zengrong Su, Dong Guo.

**Software:** Yubang Liu, Zengrong Su.

**Supervision:** Yubang Liu, Shouwen Ji, Zengrong Su.

**Validation:** Yubang Liu, Shouwen Ji.

**Visualization:** Yubang Liu.

**Writing – original draft:** Yubang Liu.

**Writing – review & editing:** Yubang Liu, Shouwen Ji.

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
