## [Decision Letter · Decision Letter 0]

17 Jun 2019

PONE-D-19-14670

Multi-objective AGV scheduling in an automatic sorting system of an unmanned (intelligent) warehouse by using two adaptive genetic algorithms and a multi-adaptive genetic algorithm

PLOS ONE

Dear Mr. Liu,

Thank you for submitting your manuscript to PLOS ONE. After careful consideration, we feel that it has merit but does not fully meet PLOS ONE’s publication criteria as it currently stands. Therefore, we invite you to submit a revised version of the manuscript that addresses the points raised during the review process.

We would appreciate receiving your revised manuscript by Aug 01 2019 11:59PM. To enhance the reproducibility of your results, we recommend that if applicable you deposit your laboratory protocols in protocols.io, where a protocol can be assigned its own identifier (DOI) such that it can be cited independently in the future. For instructions see: http://journals.plos.org/plosone/s/submission-guidelines#loc-laboratory-protocols

We look forward to receiving your revised manuscript.

Kind regards,

Yong Wang

Academic Editor

PLOS ONE

Journal Requirements:

4.   Thank you for stating the following in the Competing Interests section: "The authors have declared that no competing interests exist."

We note that one or more of the authors are employed by a commercial company:

"Beijing capital international airport Company Limited".

Additional Editor Comments (if provided):

Please see the reviewers' comments and revise your manuscript point-by-ponit according to reviewers' comments.

Reviewers' comments:

Reviewer's Responses to Questions

**Comments to the Author**

1. Is the manuscript technically sound, and do the data support the conclusions?

Reviewer #1: Yes

Reviewer #2: Yes

2. Has the statistical analysis been performed appropriately and rigorously? 

Reviewer #1: Yes

Reviewer #2: Yes

3. Have the authors made all data underlying the findings in their manuscript fully available?

Reviewer #1: Yes

Reviewer #2: No

4. Is the manuscript presented in an intelligible fashion and written in standard English?

Reviewer #1: Yes

Reviewer #2: Yes

5. Review Comments to the Author

Reviewer #1: Authors may revise this paper according to suggestions below:

1. The major contributions of this paper are not clear. Author should specifically describe them.

2. Authors may polish writing of this paper.

Reviewer #2: The research is interesting and meaningful. The makespan, the number of AGVs, and electricity consumption were taken into consideration in the scheduling process for unmanned warehouse. A multi-adaptive genetic algorithm was proposed to solve the model. The structure is clear. However, before recommending it for publication, there still need several clarifications that are listed below.

(1) In the part of Model derivation, some expressions of formulas are not standard, particularly the objective function. And equation (1) is quite confusing. I suggest explaining the equations (1)-(10), as well as other unexplained formulas, to allow a better understanding experience.

(2) Is there any criteria for choosing the weights and coefficients of the three objectives?

(3) There are lots of mistakes in the paper (e.g., some data in Table 4 is incorrect; Fig. 3 is in consistent with the description; the title of figures are not standardized), please check the whole carefully.

(4) Please put your contributions clear in the part of conclusion.

(5) The language should be improved.

(6) Please cite the following references:

Wang et al., 2017. Profit Allocation in Collaborative Multiple Centers Vehicle Routing Problem. Journal of Cleaner Production, 144:203-219.

6. PLOS authors have the option to publish the peer review history of their article (what does this mean?). If published, this will include your full peer review and any attached files.

Reviewer #1: No

Reviewer #2: No

---

## [Author Response · Author response to Decision Letter 0]

28 Jul 2019

We wish to thank you for the time and effort you have spent reviewing our paper. We are pleased to note that you have found our research work interesting and also pointed out some problems to help us improve the quality of our work.

Our itemized response to your questions, comments and suggestions:

Reviewer #1: Authors may revise this paper according to suggestions below:

1. The major contributions of this paper are not clear. Author should specifically describe them.

Response:

We have described the contribution of this paper in the conclusion section before, but it may not be prominent enough. We have rewritten the content in separate paragraphs in revised manuscript (line 473 – line 481).

The major contributions of this paper are as follows:

The work discussed in this paper can effectively improve the operation efficiency of AGV in enterprise intelligent warehouses.

The multi-adaptive genetic algorithm proposed in this paper has strong global search ability and fast optimization speed, which may help other scholars get the approximate optimal solution faster and better under the same conditions.

The scheduling model developed in this paper may inspire AGV scheduling in different research fields, such as FMS, to take speed as one of the variables and energy consumption as one of the objectives.

2. Authors may polish writing of this paper.

Response:

We have polished writing of this paper with the help of English teachers and our native English speaking friends.

Reviewer #2: The research is interesting and meaningful. The makespan, the number of AGVs, and electricity consumption were taken into consideration in the scheduling process for unmanned warehouse. A multi-adaptive genetic algorithm was proposed to solve the model. The structure is clear. However, before recommending it for publication, there still need several clarifications that are listed below.

(1) In the part of Model derivation, some expressions of formulas are not standard, particularly the objective function. And equation (1) is quite confusing. I suggest explaining the equations (1)-(10), as well as other unexplained formulas, to allow a better understanding experience.

Response:

We have rewritten this part according to the reviewer’s suggestion

We have modified the expression of makespan, adjusted the order of the equations (1)-(10), and added explanations of them in revised manuscript (line 193 – line 215).

(2) Is there any criteria for choosing the weights and coefficients of the three objectives?

Response:

The weights are determined by the degree of attention paid to the three objectives in the actual intelligent warehouse and the core idea of Pareto optimality.

The coefficients are used to adjust the values of the three objectives to have similar ranges of variation, so they are determined based on the range of variation of the three objectives.

(3) There are lots of mistakes in the paper (e.g., some data in Table 4 is incorrect; Fig. 3 is in consistent with the description; the title of figures are not standardized), please check the whole carefully.

Response:

Thank you for pointing them out. We are very sorry for our incorrect writing, and we have revised them.

(4) Please put your contributions clear in the part of conclusion.

Response:

We have rewritten our contributions in separate paragraphs in the part of conclusion (line 473 – line 481).

The major contributions of this paper are as follows:

The work discussed in this paper can effectively improve the operation efficiency of AGV in enterprise intelligent warehouses.

The multi-adaptive genetic algorithm proposed in this paper has strong global search ability and fast optimization speed, which may help other scholars get the approximate optimal solution faster and better under the same conditions.

The scheduling model developed in this paper may inspire AGV scheduling in different research fields, such as FMS, to take speed as one of the variables and energy consumption as one of the objectives.

(5) The language should be improved.

Response:

We have polished writing of this paper with the help of English teachers and our native English speaking friends.

(6) Please cite the following references:

Wang et al., 2017. Profit Allocation in Collaborative Multiple Centers Vehicle Routing Problem. Journal of Cleaner Production, 144:203-219.

Response:

We read the article carefully and found it very helpful to us - some of our revisions are based on the inspiration of this article. And we have cited it in revised manuscript(line 99 – line 101).

---

## [Decision Letter · Decision Letter 1]

19 Aug 2019

PONE-D-19-14670R1

Multi-objective AGV scheduling in an automatic sorting system of an unmanned (intelligent) warehouse by using two adaptive genetic algorithms and a multi-adaptive genetic algorithm

PLOS ONE

Dear Mr. Liu,

Thank you for submitting your manuscript to PLOS ONE. After careful consideration, we feel that it has merit but does not fully meet PLOS ONE’s publication criteria as it currently stands. Therefore, we invite you to submit a revised version of the manuscript that addresses the points raised during the review process.

We would appreciate receiving your revised manuscript by Oct 03 2019 11:59PM. To enhance the reproducibility of your results, we recommend that if applicable you deposit your laboratory protocols in protocols.io, where a protocol can be assigned its own identifier (DOI) such that it can be cited independently in the future. For instructions see: http://journals.plos.org/plosone/s/submission-guidelines#loc-laboratory-protocols

We look forward to receiving your revised manuscript.

Kind regards,

Yong Wang

Academic Editor

PLOS ONE

Additional Editor Comments (if provided):

Please see the reviewer's comments and revise your paper, thanks.

Reviewers' comments:

Reviewer's Responses to Questions

**Comments to the Author**

1. If the authors have adequately addressed your comments raised in a previous round of review and you feel that this manuscript is now acceptable for publication, you may indicate that here to bypass the “Comments to the Author” section, enter your conflict of interest statement in the “Confidential to Editor” section, and submit your "Accept" recommendation.

Reviewer #2: (No Response)

Reviewer #3: (No Response)

2. Is the manuscript technically sound, and do the data support the conclusions?

Reviewer #2: Yes

Reviewer #3: Partly

3. Has the statistical analysis been performed appropriately and rigorously? 

Reviewer #2: Yes

Reviewer #3: N/A

4. Have the authors made all data underlying the findings in their manuscript fully available?

Reviewer #2: Yes

Reviewer #3: Yes

5. Is the manuscript presented in an intelligible fashion and written in standard English?

Reviewer #2: Yes

Reviewer #3: Yes

6. Review Comments to the Author

Reviewer #2: I have no further comments. The authors have answered all my questions. This paper can be accepted.

Reviewer #3: The manuscript addresses an optimal loading model for AGV scheduling in an automatic

sorting system by using improved adaptive genetic algorithms. The topic of this paper is interesting. However, this paper has a number of major shortcomings that need to be addressed:

(1) The contributions of this paper to the literature should be clear.

(2) Authors are kindly encouraged to add some latest literature on the heuristic algorithms, optimization model of scheduling problem and vehicle routing problem.

Zhang, D., Wang, X., Li, S., Ni, N., Zhang, Z., 2018a. Joint optimization of green vehicle scheduling and routing problem with time-varying speeds. PLoS One, 1-13.

Zhang, D.Z., Zou, F.Z., Li, S.Y., Zhou, L.Y., 2017. Green Supply Chain Network Design with Economies of Scale and Environmental Concerns. J. Adv. Transp., 1-14.

Yong Wang, Xiaolei Ma, Maozeng Xu, Yinhai Wang, Yong Liu. 2015. Vehicle Routing Problem based on A Fuzzy Customer Clustering Approach for Logistics Network Optimization, Journal of Intelligent&Fuzzy Systems, 29: 1427-1442.

(3) In the section of “Algorithm design”, authors are kindly encouraged to describe clear on the repairing the infeasible solution.

(4) In the section of “Computational results and discussion “, authors are kindly encouraged to give some comparative analysis on the two proposed algorithms (i.e. AGA and MAGA)with other optimization solver ( e.g. Cplex ) on the small-size instances.

7. PLOS authors have the option to publish the peer review history of their article (what does this mean?). If published, this will include your full peer review and any attached files.

Reviewer #2: No

Reviewer #3: No

---

## [Author Response · Author response to Decision Letter 1]

17 Oct 2019

Replies to the reviewers’ comments

Reviewer #2: I have no further comments. The authors have answered all my questions. This paper can be accepted.

Response:

We appreciate your approval of our paper. Thank you for your suggestions and guidance on our article before.

Reviewer #3: The manuscript addresses an optimal loading model for AGV scheduling in an automatic sorting system by using improved adaptive genetic algorithms. The topic of this paper is interesting. However, this paper has a number of major shortcomings that need to be addressed:

(1) The contributions of this paper to the literature should be clear.

Response:

We have rewritten our contributions in a separate paragraph in the part of conclusion (line 473 – line 481).

The major contributions of this paper are as follows:

 The work discussed in this paper can effectively improve the operation efficiency of AGV in enterprise intelligent warehouses.

 The multi-adaptive genetic algorithm proposed in this paper has strong global search ability and fast optimization speed, which may help other scholars get the approximate optimal solution faster and better under the same conditions.

 The scheduling model developed in this paper may inspire AGV scheduling in different research fields, such as FMS, to take speed as one of the variables and energy consumption as one of the objectives.

(2) Authors are kindly encouraged to add some latest literature on the heuristic algorithms, optimization model of scheduling problem and vehicle routing problem.

Zhang, D., Wang, X., Li, S., Ni, N., Zhang, Z., 2018a. Joint optimization of green vehicle scheduling and routing problem with time-varying speeds. PLoS One, 1-13.

Zhang, D.Z., Zou, F.Z., Li, S.Y., Zhou, L.Y., 2017. Green Supply Chain Network Design with Economies of Scale and Environmental Concerns. J. Adv. Transp., 1-14.

Yong Wang, Xiaolei Ma, Maozeng Xu, Yinhai Wang, Yong Liu. 2015. Vehicle Routing Problem based on A Fuzzy Customer Clustering Approach for Logistics Network Optimization, Journal of Intelligent&Fuzzy Systems, 29: 1427-1442.

Response:

We read these articles carefully and we have cited them in revised manuscript(line 92 – line 94).

(3) In the section of “Algorithm design”, authors are kindly encouraged to describe clear on the repairing the infeasible solution.

Response:

We would like to explain: each solution in this paper consists of two halves, the first half is the allocation of the order sorting work, and the second half is the speed of the AGV when the order is sorted. 

If a random solution is an infeasible solution, then there should be one of two situations: battery limited or time conflict. However, the charging task is considered in the mathematical model of this paper. As long as the power of the AGV is insufficient, the charging task will be triggered: the charging time will be no longer 0, it will be calculated (line220). So, there will be no limited of battery. 

In addition, the various times (t〖ST〗_i, t〖CT〗_i,…) in this paper are calculated by the composition of the solution (line 202-207, line 216-221), so the task time of different orders sorted by the same AGV does not conflict.

So, even if it is extreme to assign all orders to the same AGV to complete the order picking, it is still a feasible solution with the longer charging time. Therefore, the solutions generated under the rules set in this paper are all feasible solutions, and there are no infeasible solutions.

(4) In the section of “Computational results and discussion “, authors are kindly encouraged to give some comparative analysis on the two proposed algorithms (i.e. AGA and MAGA)with other optimization solver ( e.g. Cplex ) on the small-size instances.

Response:

Thank you for pointing it out. In fact, at the earliest time, we really thought about doing such a comparison. Our idea at the time was to use LINGO to compare with our proposed MAGA. But after the experiment, we found that the effect of using LINGO is very unsatisfactory. Because the research in this paper not only involves the assignment of order sorting, but also the speed is variable, even if the solution space of the small-scale example reaches 1.75*10^61, it is very difficult to solve with LINGO. So in the end, we chose to use our proposed MAGA to compare with the traditional AGA proposed by others. Therefore, we hope that you understand that there is no good effect in using Cplex. We think that our algorithm (MAGA) is better than AGA, and this is enough to reflect the superior performance of the MAGA.

---

## [Editor Report · Decision Letter 2]

21 Nov 2019

Multi-objective AGV scheduling in an automatic sorting system of an unmanned (intelligent) warehouse by using two adaptive genetic algorithms and a multi-adaptive genetic algorithm

PONE-D-19-14670R2

Dear Dr. Liu,

We are pleased to inform you that your manuscript has been judged scientifically suitable for publication and will be formally accepted for publication once it complies with all outstanding technical requirements.

With kind regards,

Yong Wang

Academic Editor

PLOS ONE
---

## [Editor Report · Acceptance letter]

27 Nov 2019

PONE-D-19-14670R2 

Multi-objective AGV scheduling in an automatic sorting system of an unmanned (intelligent) warehouse by using two adaptive genetic algorithms and a multi-adaptive genetic algorithm 

Dear Dr. Liu:

I am pleased to inform you that your manuscript has been deemed suitable for publication in PLOS ONE. Congratulations! Your manuscript is now with our production department. 

With kind regards,

on behalf of

Dr. Yong Wang 

Academic Editor

PLOS ONE